# Association of miRNA and mRNA Levels of the Clinical Onset of Multiple Sclerosis Patients

**DOI:** 10.3390/biology10060554

**Published:** 2021-06-20

**Authors:** Danuta Piotrzkowska, Elzbieta Miller, Ewa Kucharska, Marta Niwald, Ireneusz Majsterek

**Affiliations:** 1Department of Chemistry and Clinical Biochemistry, Medical University of Lodz, 90-136 Lodz, Poland; danuta.piotrzkowska@umed.lodz.pl; 2Department of Neurological Rehabilitation, Medical University of Lodz, Milionowa 14, 93-113 Lodz, Poland; elzbieta.dorota.miller@umed.lodz.pl (E.M.); marta.niwald@umed.lodz.pl (M.N.); 3Department Geriatrics and Social Work, Jesuit University Ignatianum in Cracow, Kopernika 26, 31-501 Krakow, Poland; ewa.kucharska@ignatianum.edu.pl

**Keywords:** neurodegeneration, multiple sclerosis, neurotrophins, heat shock proteins, microRNAs

## Abstract

**Simple Summary:**

In this study, we investigated the effect of microRNAs on the expression level of neuroprotective proteins, heat shock proteins, and sirtuin in peripheral blood mononuclear cells in the development of multiple sclerosis. Our results show that the gene expression of neurotrophins, heat shock proteins, SIRT1, and miRNAs by the immune cells of MS is d changed. A decrease in the expression of the BDNF and SIRT1 genes and an increase in the expression of miR-132-3p, miR-34a, and miR-132 in PBMCs may indicate an inhibition of the neuroprotective function of these cells, which may be associated with the transition of the immune system towards inflammation in the development of multiple sclerosis.

**Abstract:**

Multiple sclerosis (MS) is a demyelinating disease characterized by chronic inflammation of the central nervous system, in which many factors can act together to influence disease susceptibility and progression. To date, the exact cause of MS is still unclear, but it is believed to result from an abnormal response of the immune system to one or more myelin antigens that develops in genetically susceptible individuals after their exposure to a, as yet undefined, causal agent. In our study, we assessed the effect of microRNAs on the expression level of neuroprotective proteins, including neurotrophins (BDNF and NT4/5), heat shock proteins (HSP70 and HSP27), and sirtuin (SIRT1) in peripheral blood mononuclear cells in the development of multiple sclerosis. The analysis of dysregulation of miRNA levels and the resulting changes in target mRNA/protein expression levels could contribute to a better understanding of the etiology of multiple sclerosis, as well as new alternative methods of diagnosis and treatment of this disease. The aim of this study was to find a link between neurotrophins (BDNF and NT4), SIRT1, heat shock proteins (HSP27 and HSP27), and miRNAs that are involved in the development of multiple sclerosis. The analysis of the selected miRNAs showed a negative correlation of SIRT1 with miR-132 and miR-34a and of BDNF with 132-3p in PBMCs, which suggests that the miRNAs we selected may regulate the expression level of the studied genes.

## 1. Introduction

Multiple sclerosis (MS) is a serious neurological disorder affecting young Caucasian individuals, especially women, usually with an age of onset at 18–40 years. Four clinical forms can be distinguished based on the clinical onset and disease progression: relapsing-remitting (RRMS), the most frequent clinical form affecting approximately 80–85% of MS patients, secondary progressive (SPMS), primary progressive (PPMS), and progressive relapsing (PRMS). MS is not inherited, however, it has been shown that the probability of multiple sclerosis in a family member is higher (0.8–3% depending on the nature of the relationship with the patient and 25% in the case of identical twins) than in subjects without the MS history in their families (0.001%) [1]. MS affects more than 2.3 million people worldwide. The etiology and pathogenesis of MS remain elusive, but it is believed to result from an abnormal response of the immune system to one or more myelin antigens that develop in genetically susceptible individuals after their exposure to an as-yet undefined causal agent [2].

Clinical changes include multifocal defects within the white matter of the brain and spinal cord, which ultimately lead to the destruction of the myelin sheath of nerve fibers (demyelination) and the damage to glial cells (oligodendrocytes). Oligodendrocytes are the main source of myelin in central nervous system. Myelin provides adequate axonal isolation and is essential for the proper conduction of nerve stimuli [3].

The clinical course of the disease, as well as symptoms, can vary. Typically, in the course of MS, periods of exacerbation of the disease are observed (relapsing), in which the neurological symptoms appear in various combinations. Usually, after some time, the symptoms subside completely or partially and remissions occur, during which the relative disease stabilization is observed [4].

In multiple sclerosis, miRNAs profile has been observed to be changed within the CNS and the immune system, which results in the altered levels of gene expression in many cell types involved in the disease [5]. Numerous studies have demonstrated that miRNA expression is dysregulated in MS patients both in blood cells and CNS lesions. The CNS tissue analysis has revealed a signature of 50 miRNAs that are up-regulated and 30 miRNAs that are down-regulated in MS compared to healthy subjects [6,7]. In whole blood, 165 miRNAs were reported to be significantly up- or down-regulated [8]. Furthermore, differential expression of various miRNAs in peripheral blood mononuclear cells of MS patients has been identified during both relapses and remissions [6]. However, there are no data about the role of miRNAs in the regulation of the expression level of neuroprotective proteins, including neurotrophies, heat shock proteins, and sirtuins, in the development of multiple sclerosis.

MicroRNAs (miRNAs) are a family of small non-coding RNAs of 18–25 nucleotides that are transcribed from DNA sequences into primary miRNAs and then processed into precursor and mature miRNAs. The perfect or near-perfect complementary binding of miRNAs to their target mRNAs negatively regulates gene expression in terms of accelerating mRNA degradation or suppressing mRNA translation [9]. MiRNAs are found in all body fluids and tissues and in most cell types [10]. Each miRNA is able to regulate the expression of several (maybe even hundreds of) target genes and is involved in important processes, such as the embryonic development, immune response, inflammation, oncogenesis, as well as cellular growth and proliferation [11].

Therefore, analysis of miRNA dysregulation and the resulting changes in the mRNA/protein expression level may contribute to a better understanding of the multiple sclerosis etiology as well as new alternative methods for the diagnosis and treatment of the disease [12].

In our study, we assessed the effect of miRNAs on the expression level of neuroprotective mRNA, including neurotrophins (BDNF and NT4/5), heat shock proteins (HSP70 and HSP27), and sirtuin (SIRT1) in peripheral blood mononuclear cells in the development of MS. The aim of this study was to find a link between neurotrophins (BDNF and NT4/5), SIRT1, and heat shock proteins (HSP27 and HSP27), as well as miRNAs that are involved in the development of MS.

Using software (miRNApath and miRTarBase) and literature data, we selected 12 miRNAs that can regulate the expression of BDNF genes (miR-132, miR-182-5p, and miR-134), NT4/5 (miR-21-5p), SIRT1 (miR-132-3p, miR-34a, and miR-34c), HSP70 (miR-378-3p and miR-181b-5p), and HSP27 (miR-577 and miR-17-5p).

In neurodegenerative diseases such as MS, there is an increased need for neurotrophic factors influencing neuroplasticity and neuroprotection in the CNS. In MS patients, the influx of peripheral blood mononuclear cells (PBMC) to the CNS can compensate for the deficiency of neuroprotective factors, in particular, nerve growth factor (NGF), brain-derived neurotrophic factor (BDNF), neurotrophin 3 (NT-3), and neurotrophin 4/5 (NT-4/5), reducing the rate of cerebral atrophy. So far, a relationship has been demonstrated between the BDNF factor secreted by PBMC and the protective effect on neuronal axons in the course of MS [5].

Heat shock proteins (HSPs) are a group of phylogenetically conserved proteins found in all prokaryotic and eukaryotic cells. These proteins are named according to their molecular weight, which ranges from 17 to more than 100 kDa [13].

HSPs have been shown to play a neuroprotective role not only by preventing aggregation of misfolded proteins but also by inducing anti-apoptotic mechanisms.

The inflammatory process observed in the early stages of MS acts as a stimulus for the induction of HSPs expression by glial cells and their subsequent release into the extracellular environment, which protects nerve cells during subsequent neurodegenerative phases of the disease. In lesions within the CNS in patients with multiple sclerosis and in an animal model of MS disease, overexpression of most HSPs induced by inflammation and oxidative stress was observed [14,15].

Therefore, disturbances in HSPs synthesis may lead to rapid progression of degenerative changes.

Sirtuins (SIRTs) are a group of histone deacetylases whose activities are dependent on and regulated by nicotinamide adenine dinucleotide (NAD+).

SIRTs stimulate expression of genes responsible for energy metabolism and pro-survival mechanisms. Additionally, SIRT1 selectively suppresses genes involved in apoptosis and inflammation. Therefore, sirtuins play a role in multifaceted mechanisms that lead to increased cell viability. SIRT1 is the best known sirtuin among the seven. In an animal model of multiple sclerosis, SIRT1 activation by SRT501 prevents neuronal loss and improves neuronal dysfunction [4].

The main purpose of this work is to assess the expression of genes encoding proteins involved in the neuroprotection in MS. Failure to produce these proteins might result in decreased protection of the central nervous system, consequently leading to increased atrophy, which is the main determinant of MS patients’ end-point disability. Results from both clinical research and animal models suggest that these factors play a pivotal role in neuroprotective and neuroregenerative processes that are often defective in the course of MS. However, there are no data about role of microRNAs in regulation of expression level of neuroprotective protein, including neurotrophins, heat shock proteins, and sirtuins in multiple sclerosis development.

In our study, we assessed the effect of miRNAs on the expression level of neuropro-tective proteins, including neurotrophins (BDNF and NT4/5), heat shock proteins (Hsp70 and HSP27), and SIRT1 in the development of multiple sclerosis. We postulate that dysregulation of miRNA levels and the resulting changes in target mRNA/protein expression levels could contribute to development of multiple sclerosis.

## 2. Materials and Methods

### 2.1. Materials

Twenty-eight patients, 18 females and 10 males (aged 59.9 ± 5.5 years), 13 with diagnosed relapsing-remitting multiple sclerosis (RRMS) and 15 with secondary progressive multiple sclerosis (SPMS) were hospitalized at the General Hospital of the Third Department of Neurological Rehabilitation in Lodz. RR-MS patients were in the remission (stable) phase, without attacks or steroid treatment for over 2 years. All MS patients had not received any immunomodulatory therapy within 3 months prior to blood withdrawal. The control group included 33 subjects, 18 females and 15 males, aged 55.5 ± 7.2 years (Table 1). All patients and control subjects enrolled to the study were Caucasians. The study design was approved by the Committee for Bioethics of the Medical University of Lodz (Poland) and met the principles of the Declaration of Helsinki. An informed written consent was explained to and signed by all participants before the study was initiated. MS was clinically diagnosed in patients according to the McDonald’s criteria. Exclusion criteria were as follows: age below 18 years and above 70 years, severe general condition of the patient, the presence of another neurological disease, the presence of another autoimmune disease, the presence of cancer, difficult logical verbal contact with the patient (preventing the signing of informed consent to the study), severe psychiatric illness (preventing the signing of informed consent to the study), and active inflammatory acute disease. Prior to the examination, multiple sclerosis patients did not show any additional inflammatory disease or cancer at the time of blood collection.

### 2.2. Methods

#### 2.2.1. Blood Sample Collection, RNA Preparation, and Reverse Transcription Quantitative Real-Time PCR RNA. Isolation and cDNA Synthesis

Nine milliliters of blood was drawn by venepuncture into a citrate phosphate dextrose adenine contained in a vacuum tube (Vacuette), which was kept in a fridge after the blood withdrawal. Total RNA was isolated from PBMC using a Fenozol reagent and Total RNA Mini Plus (A&A Biotechnology), according to the manufacturer’s protocol. Total RNA was extracted from 28 patients with MS and 33 subjects without MS. RNA was eluted in 50 μL RNase-free water and stored at −20 °C. RNA quality and quantification were measured spectrophotometrically using a Synergy/HT spectrophotometer and software, applying the 260/280 and 260/230 ratio algorithms. RNA with a 260/280 nm ratio in the range of 1.8–2.0 was considered high quality and was used for further analysis. cDNA was synthesized from 1 μg of total RNA with a High Capacity cDNA Reverse Transcription Kit (Life Technologies, Carlsbad, CA, USA) following the manufacturer’s protocol. The cDNA was subjected to quantitative real-time PCR using the CFX Connect Real-Time System (BioRad, Hercules, CA, USA) with TaqMan PCR Master Mix and TaqMan Gene Expression Assays (Applied Biosystems, Waltham, MA, USA) for BDNF, NT4/5, SIRT1, HSP70, HSP27, and GAPDH mRNA (BDNF Hs02718934_s1, NT4/5 Hs01921834_s1, SIRT1 Hs01009005_m1, HSP70 Hs04187663_g1, HSP27 Hs03044127_g1, and GAPDH Hs03929097_g1), which were used according to the manufacturer’s instruction. The GAPDH gene was used as the internal sample control. Briefly, 50 ng cDNA was added to a single reaction. All samples were analyzed in duplicate, and in the event of a discrepancy of results (Ct), the result was rejected and the level of expression of a given gene in a given patient was analyzed again. A positive result was defined by a threshold cycle (Ct) value lower than 35 (the Ct value is determined by the number of cycles needed to exceed the background signal). Abundance of mRNAs in the studied material was quantified by the 2^-ΔCt^ method.

For measuring miRNA expression levels, cDNA samples were prepared from 350 ng RNA samples using the High Capacity cDNA Reverse Transcription Kit (Applied Biosystems, Waltham, MA, USA) in accordance with the protocol for Creating Custom RT Pools using TaqMan ^®^ MicroRNA Assays from Applied Biosystems. Real-time PCR reactions were performed with 10 μL mixtures containing 10 ng of cDNA, TaqMan^®^ Fast Advanced Master Mix (Applied Biosystems, Waltham, MA, USA), and Primers (TaqMan™ MicroRNA Assay; Applied Biosystems). U6 snRNA, miR-103, and miR-191 were used as reference genes as recommended for the transcriptomic analysis. TaqMan^®^ MicroRNA Assay IDs are listed below: hsa-miR-132-3p-002132, hsa-miR-182-5p-002334, mmu-miR-134-001186, hsa-miR-221-002096, hsa-miR-21-5p-000397, hsa-miR-378a-3p-000567, hsa-miR-181b-001098, hsa-miR-577-002675, hsa-miR-17-17p-002308, hsa-miR-34a-000426, hsa-miR-132-000457, hsa-miR-34c-000428, TaqMan™ microRNA Control Assay, RNU6B-001973, has-miR-191-5p-002299, and hsa-miR-103a-3p-000439.

#### 2.2.2. Statistical Analysis

All statistical analyses were conducted with Prism 8 (GraphPad Software, San Diego, CA, USA). The data are presented as the means ± SEM of two or three independent experiments performed in two or more repetitions. Distribution of variables was assessed by the Shapiro–Wilk test. Statistical analysis of differences between the groups of data was carried out using the Mann–Whitney U-test (for non-normal distribution). Statistical analysis of differences between the groups of data was carried out using the Mann–Whitney U-test. Values of *p* < 0.05 were regarded as statistically significant (*p** ≤ 0.05, *p*** ≤ 0.01, and *p***** ≤ 0.0001). The correlation analysis was performed by the Spearman correlation. *p*-values < 0.05 were considered significant.

## 3. Results

### 3.1. Expression Levels of Gene Neurotrophins (BDNF and NT4/5), Heat Shock Protein HSP (HSP70 and HSP27), SIRT1, and miRNAs in PBMCs in Patients with Multiple Sclerosis

We checked the expression level of genes, BDNF, NT4/5, HSP70, HSP27, and SIRT1 in PBMCs in patients with multiple sclerosis (patients with mild to moderate disability) and in the control group, healthy adults. Using software and literature data, we selected 12 miRNAs that can regulate the expression of BDNF genes (miR-132, miR-182-5p, and miR-134), NT4/5 (miR-21-5p), SIRT1 (miR-132-3p, miR-34a, and miR-34c), HSP70 (miR-378-3p and miR-181b-5p), and HSP27 (miR-577 and miR-17-5p). The analysis of the expression level of the studied genes and miRNAs was performed on the same group of patients, thanks to which we could observe direct interactions between the tested miRNAs and the target gene.

The analysis of the expression level of miRNAs in PBMCs in MS patients in comparison with the control group showed changes in the expression level in 6 out of 12 selected miRNAs. In the graphs, we showed only those miRNAs whose expression level differed in MS patients compared to the control at the significance level of *p* ≤ 0.05.

#### 3.1.1. Expression Levels of Neurotrophins (BDNF and NT4/5) and miRNAs in PBMCs of Patients with Multiple Sclerosis

Patients with multiple sclerosis showed a lower level of BDNF gene expression in PBMCs than that in the healthy group. The BDNF gene expression in patients with multiple sclerosis showed a 2-fold decrease compared to the control group (*p* < 0.005) (Figure 1a). In the same patients, we analyzed the expression level of miR-132, miR-182-5p, and miR-134 in PBMC cells. We found a 2-fold increase in the level of the miR-132 expression and a decrease in the level of the miR-134 expression in MS patients compared to the control group (Figure 1b,c). We obtained the statistical significance for miR-132 and miR-134 of *p* = 0.0398 and *p* = 0.0289, respectively. The level of miR-182-5p expression (Figure 1d) in patients with MS was elevated but we did not show the statistical significance. The level of the miR-132 expression negatively correlated with the level of BDNF (R = −0.3358 *p* = 0.001) (Figure 1e), while the level of miR-134 positively correlated with the level of BDNF (R = 0.5535, *p* ≤ 0.0001) (these data were combined for all data *N* = 61) (Figure 1f).

The analysis of the NT4/5 gene expression in PBMCs showed an increase in the NT4/5 mRNA level in the group of patients with multiple sclerosis (*N* = 25) compared to the control group (*N* = 33) (Figure 2a). Patients with multiple sclerosis revealed a 2-fold increase in the NT4/5 gene expression compared to the control group (*p* < 0.05). The analysis of the miR-21-5p expression level in PBMCs indicated a 1.4-fold increase in the miR-21-5p level in MS patients compared to the control group, *p* = 0.0185 (Figure 2b). No correlation was observed between the level of the NT4/5 expression and the level of miR-21-5p. The analysis of miR-21-5p levels in PBMCs was performed according to the type of multiple sclerosis: RRMS (*N* = 13) and SPMS (*N* = 15). The analysis of the expression level showed that patients with RRMS disease type (*N* = 13) had a 1.5-fold higher expression level of miR-21-5p compared to the SPMS group (*N* = 15) (Figure 2c).

#### 3.1.2. Expression Levels of Heat Shock Proteins HSP27 and HSP70 and miRNAs in PBMCs of Patients with Multiple Sclerosis

The analysis of the mRNA levels of the heat shock proteins HSP27 and HSP70 in PBMC in patients with multiple sclerosis and in the control group showed the changes in the expression levels of both genes. We demonstrated a 2-fold reduction in the level of the HSP27 gene expression in PBMCs in MS patients (*N* = 28) compared to the control group (*N* = 33), *p* < 0.0001 (Figure 3a). In contrast, the level of the HSP70 expression exhibited a 2-fold increase in the group of patients with multiple sclerosis (*N* = 28) than in the control group (*N* = 33), *p* < 0.05 (Figure 3b). Next, the levels of the expression of selected miRNAs which can regulate the expression of HSP70 (miR-378, miR-181b-5p) and HSP27 (miR-577, miR-17) were analyzed. In the case of miR-181b-5p, a 2-fold increase in their expression level in MS patients (*N* = 28) was observed compared to the control group (*N* = 33), (*p* < 0.005) (Figure 3c), while miR-378, miR-577, and miR-17 showed no significant changes in their expression level in MS patients compared to the control group (data not shown). The level of miR-181b-5p correlated positively with the level of HSP70 (R = 0.4389 *p* = 0.0005), and these data were combined for all data (*N* = 61) (Figure 3e). The analysis of miR-181b-5p levels in PBMCs was performed according to the type of multiple sclerosis: RRMS (*N* = 13) and SPMS (*N* = 15) showed no significant differences (Figure 3d).

#### 3.1.3. Expression Levels of SIRT1 and miRNAs in PBMCs of Patients with Multiple Sclerosis

In patients with MS, a 1.8-fold decrease in the level of the SIRT1 gene expression (*N* = 28) was observed compared to the control group (*N* = 33) (*p* < 0.05) (Figure 4a). Next, the expression level of miR-34a, miR-34c, and miR-132-3p in PBMCs in MS patients (*N* = 28) and control group (*N* = 33) was analyzed. The analysis of the expression level of miRNAs in PBMCs showed a 1.5-fold increase in miR-34a (*p* = 0.0042) (Figure 4b) and increase in miR-132-3p (*p* = 0.0138) levels in MS patients compared to the control group (Figure 4c). The analysis of the expression level of miR-34c in patients with MS revealed an increase in the expression level, however, we did not reach the statistical significance (data not shown). The level of miR-34a and miR-132-3p negatively correlated with the level of SIRT1 (R = −0.4897, *p* = 0.0006) (Figure 4d) (R = −0.6604) (*p* < 0.0001) (Figure 4e); these data were combined for all data (*N* = 61).

## 4. Discussion

In our study, we assessed the effect of miRNAs on the expression level of neuroprotective proteins, including neurotrophins (BDNF and NT4/5), heat shock proteins (Hsp70 and HSP27), and SIRT1 in the development of multiple sclerosis. The analysis of dysregulation of miRNA levels and the resulting changes in target mRNA/protein expression levels could contribute to a better understanding of the etiology of multiple sclerosis, as well as to new alternative methods of diagnosis and treatment of this disease. The aim of this study was to find out a link between neurotrophins (BDNF and NT4/5), SIRT1, heat shock proteins (HSP70 and HSP27), and miRNAs that are involved in the development of multiple sclerosis. We selected several miRNAs for each of the studied genes: BDNF (hsa-miR-132-3p, hsa-mir-182-5p, and hsa-miR-134), NT-4/5 (hsa-miR-21-5p), NGF (hsa-miR-221), HSP70 (hsa-mir-378a-3p and hsa-mir-181b-5p), HSP27 (hsa-miR-577 and hsa-miR-17-5p), and SIRT1 (hsa-miR-34a, hsa-miR-132, and hsa-miR-34c). When selecting miRNAs, we were guided by bioinformatic research and literature data. Each miRNA was tested in the miRNA target software (miRNApath and miRTarBase) to assess complementarity of the tested gene sequence. We checked the mRNA levels of neurotrophins, heat shock proteins, sirtuin 1, and miRNAs in peripheral blood mononuclear cells of RRMS and SPMR patients (patients with mild to moderate disabilities) and healthy controls. Multiple sclerosis patients (both RRMS and SPMS patients) had a mild to moderate disability.

Our analysis showed a decrease in the expression of the BDNF gene in MS patients compared to the control group and a simultaneous increased level of miR-132 and miR-182-5p expression. Our results are consistent with the study by You-Jie Li, who noticed that transfection of neuronal cells (SH-SY5Y) with miR-132 and mi-182-5p resulted in a reduction in BDNF expression. Moreover, studies conducted on patients with depression also showed a decrease in BDNF expression (ELISA) and a higher level of miR-132 and miR-182-5p expression in the serum (real-time PCR method) [16]. These results confirm our observation that in multiple sclerosis, we can observe the influence of miRNA-132 and miR-182-5p interference on the expression of the BDNF gene.

In the case of miR-134, we showed a positive correlation between the level of BDNF expression and the level of miR-134 in PBMC. Some literature data say that it is BDNF that influences the expression of miRNA-134 by activating the TrkB pathway [17], which would confirm our results, where, with low levels of BDNF expression, we observed a reduction levels of miR-134 from MS patients.

The next neurotrophin whose expression level was analyzed in PBMC in MS and control patients was NT4/5. Analysis of the expression level of NT4/5 neurotrphine showed a 2-fold increase in expression in MS patients compared to the control group. In contrast, expression of miR-21-5p, which was selected as miRNA that can regulate NT4/5 levels, showed a 1.5-fold increase in the expression level in PBMC in MS patients compared to the control group. Additionally, we found that the expression level of miR-21-5p in PBMC depends on the type of multiple sclerosis. Patients with RRMS type had 1.5 times higher level of miR-21-5p expression compared to patients with SPMS type. The observed increase in NT4/5 and miR-21-5p expression levels can be explained by the immune system’s response to neurodegenerative changes and inflammatory factors in MS patients. Numerous studies suggest that miR-21-5p plays a key role in the production of pro-inflammatory cytokines, which may be a contributing factor in the development of autoimmune diseases such as multiple sclerosis. MiR-21 has been documented to stimulate the production of IFN-γ and IL-17A in T cells, which are responsible for the development of Th1 and Th17. The main pathological action of Th17 is the maintenance of chronic inflammation, which can be seen in MS [18,19].

The next proteins we analyzed were HSP70 and HSP27. Intracellular HSP70 mediates chaperone-cytoprotective functions as well as can block multiple steps in the apoptotic pathway. In contrast, extracellular HSP70 promotes innate and adaptive immune responses. Thus, HSP70 can be considered a critical molecule in the pathogenesis of multiple sclerosis and a potential target in this disease due to its immune and cytoprotective functions [20]. In our study, we checked the expression level of the HSP70 genes in PBMCs from MS patients and the control group.

Early studies by Lechner confirm our observation, he showed that the expression level of HSP70 in MS patients is increased compared to control patients, however, he found that HSP70 expression in patients with other inflammatory neurological diseases (OIND) is higher than in MS. This finding can be explained by the fact that inflammation in MS is generally less pronounced compared to other inflammatory diseases of the CNS [20]. Other studies have shown that the presence of HSP70 is required for the proper functioning of miRNAs that promote a pro-inflammatory effect through Th17. It was shown that downregulation of HSP70 led to a significant suppression of the expression of Th17 marker genes. Therefore, we believe that intracellular levels of HSP70 may be a good indicator of a state of stress, and inhibition of HSP70 may help ameliorate pro-immune responsiveness. Experiments in animal models of EAE (Experimental Autoimmune Encephalomyelitis) have shown that lowering HSP70 protein levels can combat the pro-inflammatory effect via Th17 [20].

Another HSP family protein that we analyzed was the HSP27 protein. This protein belongs to the small subfamily of HSP and plays an important role in the inhibition of apoptosis. It has been shown that the small heat shock protein HSP27 has a stronger protective effect in the nervous system [20]. Therefore, we decided to investigate the level of HSP27 gene expression in PBMC of MS patients and control patients.

The results obtained by us may indicate that we do not observe any neuroprotective effects on the part of immune cells in patients with MS. On the other hand, it should be noted that the HSP27 protein is an anti-apoptotic protein, and its overexpression is detected in tumors of the central nervous system, breast, ovary, lung and skin [21,22].

Expression analysis of the selected miRNAs for HSPs showed changes in expression only in miR-181b-5p, which was selected as the regulator of HSP70 gene expression. We observed a 2-fold increase in the expression level of miR-181b-5p in PBMC from MS patients, and the level of miR-181-5b was positively correlated with the level of HSP70.

MiR-181-5p regulates many important biological processes such as cell proliferation, apoptosis, autophagy, mitochondrial function, and immune response. Importantly, several studies have shown abnormal expression of these miRNAs in neurodegenerative disorders.

SIRT1 selectively inhibits genes involved in apoptosis and inflammation [23]. Recently, many reports indicate the neuroprotective role of SIRT1 in both acute and chronic neurological diseases. Therefore, one of the main goals of this study was to evaluate the expression of genes encoding proteins involved in neuroprotection in MS. The analysis of SIRT1 gene expression levels in PBMCs in MS patients showed an almost 1.8-fold reduction compared to the control group. In contrast, the level expression of selected miRNAs: hsa-miR-34a and hsa-miR-132-3p increased. The level of miR-34a and miR-132-3p negatively correlated with the level of SIRT1. The results demonstrated that the miR-34a expression was significantly increased in RRMS patients compared to those with SPMS and the control group. Previously, it was shown that the level of miR-34a expression is upregulated in active lesions in MS patients [6]. This result, along with ours, confirms the possible contribution of miR-34a to the suppression of Treg formation and the bias towards the development of Th17 and the development of the inflammatory process. Because many studies in animal models of demyelinating and neurodegenerative diseases have shown that SIRT1 induction can ameliorate the course of the disease. SIRT1 represents a possible biomarker of relapses and a potential new target for therapeutic intervention in MS. Modulation of SIRT1 may be a valuable strategy for treating or preventing MS as well as neurodegenerative disorders of the central nervous system [23,24].

Based on the conducted research, we can conclude that the gene and microRNA expression profiling may be a good diagnostic tool in the future for assessing the severity of the disease or estimating survival time, and it may also be helpful in choosing a treatment tailored to the individual patient’s needs.

## 5. Conclusions

Our results show that the gene expression of neurotrophins, heat shock proteins, SIRT1, and miRNAs by the immune cells of MS is impaired. A decrease in the expression of the BDNF and SIRT1 genes and an increase in the expression of miR-132-3p, miR-34a, and miR-132 in PBMCs may indicate an inhibition of the neuroprotective function of these cells, which may be associated with the transition of the immune system towards inflammation by producing IFN-γ and IL-17A in T cells, responsible for the development of Th1 and Th17, and, consequently, by maintaining the chronic inflammation that can be seen in MS.

In conclusion, our results suggest that the levels of mRNA of the selected neurotrophins (BDNF and NT4/5), heat shock proteins (HSP70 and HSP27), SIRT1, and selected miRNAs in peripheral blood mononuclear cells may serve as biomarkers of inflammation in the CNS and neurodegenerative processes in patients with multiple sclerosis.

## Figures and Tables

**Figure 1 biology-10-00554-f001:**
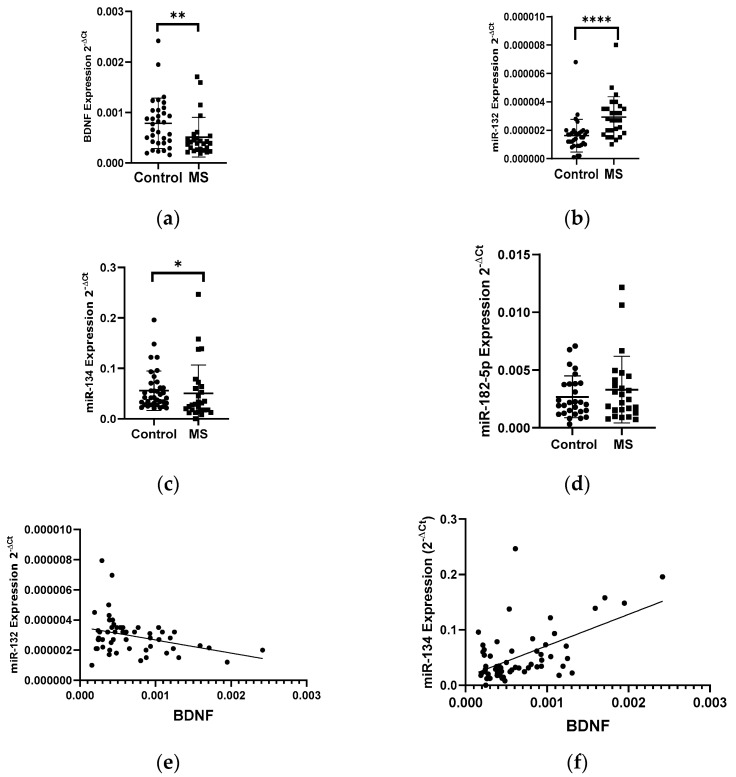
Expression levels and correlations of BDNF, miR-132, and miR-134 in PBMCs from multiple sclerosis patients and healthy controls. Expression levels 2^−ΔCt^ are shown for: (**a**) BDNF, (**b**) miR-132, (**c**) miR-134, and (**d**) miR-182-5p. (**e**,**f**) Spearman correlation (R) plots for the expression levels of BDNF with miR-132 and with miR-134 (combined for all cohorts; *N* = 61). The PBMCs were from multiple sclerosis patients (*N* = 27), aged 59 ± 5.5 years, and healthy controls, aged 60 ± 6 years (*N* = 33). Expression levels were determined by real-time PCR (see Methods). Statistical analysis of differences between the groups of data was carried out using the Mann–Whitney U-test. Values of *p* ≤ 0.05 were regarded as statistically significant (*p** ≤ 0.05, *p*** ≤ 0.01, and *p***** ≤ 0.0001).

**Figure 2 biology-10-00554-f002:**
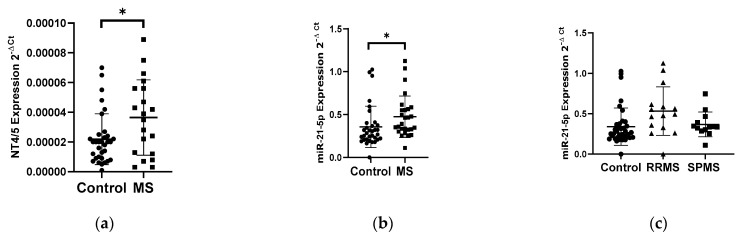
Expression levels of NT4/5 and miR-21-5p in PBMCs from multiple sclerosis patients and healthy controls. Expression levels (2^−ΔCt^) are shown for: (**a**) NT4/5 and (**b**) miR-21-5p, and (**c**) the expression level of miR-21-5p in PBMCs of MS patients according to type: RRMS (*n* = 13), SPMS (*N* = 15), and the control group (*N* = 33). The PBMCs were from multiple sclerosis patients (*N* = 28), aged 59 ± 5.5 years, and healthy controls, aged 60 ± 6 years (*N* = 33). Expression levels were determined by real-time PCR (see Methods). Statistical analysis of differences between the groups of data was carried out using the Mann–Whitney U-test. Values of *p* ≤ 0.05 were regarded as statistically significant (*p** ≤ 0.05).

**Figure 3 biology-10-00554-f003:**
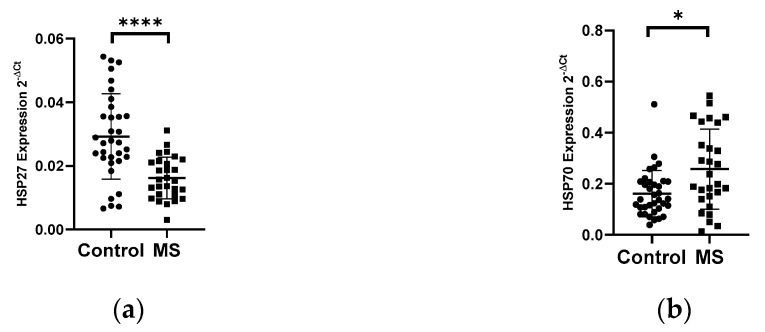
Expression levels and correlations of HSP27 and HSP70 and miR-18b1-5p in PBMCs from multiple sclerosis patients and healthy controls. Expression levels (2^−ΔCt^) are shown for: (**a**) HSP27; (**b**) HSP70, and (**c**) miR-181b-5p, and (**d**) the expression level of miR-181-5p in PBMCs of MS patients according to type: RRMS (*n* = 13) and SPMS (*N* = 15). (**e**) Spearman correlation (R) plots for the expression levels of HSP70 with miR-181b (combined for all cohorts; *N* = 60). The PBMCs were from multiple sclerosis patients (*N* = 28), aged 59 ± 5.5 years, and healthy controls, aged 60 ± 6 years (*N* = 33). Expression levels were determined by real-time PCR (see Methods). Statistical analysis of differences between the groups of data was carried out using the Mann–Whitney U-test. Values of *p* ≤ 0.05 were regarded as statistically significant (*p** ≤ 0.05, *p*** ≤ 0.01, and *p***** ≤ 0.0001).

**Figure 4 biology-10-00554-f004:**
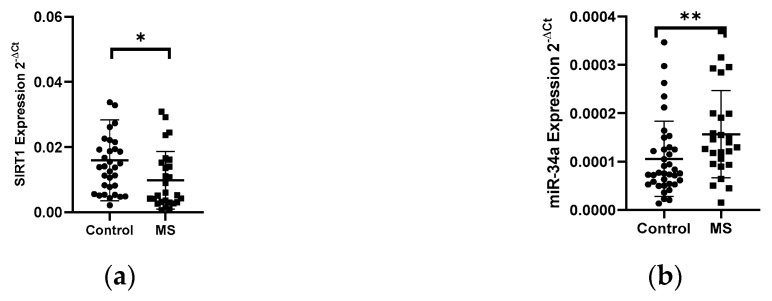
Expression levels and correlations of SIRT1, miR-34a, and miR-132-3p in PBMCs from multiple sclerosis patients and healthy controls. Expression levels (2^−ΔCt^) are shown for: (**a**) SIRT1, (**b**) miR-34a, and (**c**) miR-132-3p. (**d**,**e**) Spearman correlation (R) plots for the expression levels of SIRT1 with miR-34a and miR-132-3p (combined for all cohorts; *N* = 61). The PBMCs were from multiple sclerosis patients (*N* = 27). Expression levels were determined by real-time PCR (see Methods). Statistical analysis of differences between the groups of data was carried out using the Mann–Whitney U-test. Values of *p* ≤ 0.05 were regarded as statistically significant (*p** ≤ 0.05, *p*** ≤ 0.01).

**Table 1 biology-10-00554-t001:** Clinical characteristics of the 13 patients with relapsing-remitting multiple sclerosis (RRMS) and 15 patients with secondary progressive multiple sclerosis (SPMS).

	Total
Number of patients	28
RRMS/SPMS	13/15
Females/males	18/10
Age (years, mean +/− SD)	59.5 ± 5.5
Disease duration (years, mean +/− SD)	9.7 ± 3.2
Expanded disability status scale (EDSS) at the stable phase (range)	6 ± 1.0

## Data Availability

All data used to support the findings of this study are included within the article.

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
