# Peer review of "Association of miRNA and mRNA Levels of the Clinical Onset of Multiple Sclerosis Patients"

_biology, 2021, doi:10.3390/biology10060554_

Round 1

Reviewer 1 Report

The authors present interesting work on the association of miRNA and mRNA levels of the clinical onset of 2 multiple sclerosis patients. 

The entire manuscript should be carefully edited.

minor points:

Line 50 rewrite

Line 82 microRNA should be indicated earlier in the text

Line 94-95 rewrite

Author Response

Response to Reviewer 1 Comments

 Point 1:

Line 50 rewrite

Clinical changes include multifocal defects within the white matter of the brain and spinal cord, which ultimately lead to the destruction of the myelin sheath of nerve fibers (demyelination) and the damage to glial cells (oligodendrocytes). Oligodendrocytes are the main source of myelin in central nervous system. Myelin provides adequate axonal isolation and is essential for the proper conduction of nerve stimuli [3].

Point 2:

Line 82 microRNA should be indicated earlier in the text

The entire paragraph (lines 82-90) should be earlier on line 69

Line 70

Point 3:

Line 94-95 rewrite

During the development of the nervous system, neurotrophin deprivation serves as a physiological mechanism of neuronal elimination. In the adult CNS, neurotrophins play a protective role towards specific neuronal populations [13].

Introdactions have been shortened and this paragraph has been removed.

Best regards

D.Piotrzkowska

Reviewer 2 Report

Overall the data is reasonably well presented, the standard of English is good and the methodology used is clear. However, there are some issues with the manuscript that should be addressed.

The Introduction is too long.

2.2.2. Statistical Analysis: The information  if the distribution is normal or not  (probably not because these are human samples) should be included.

Line 225-227, last sentence- should be included to Figure legend.

Line 327-329, authors write: In our study, we assessed the effect of microRNAs on the expression level of neuroprotective proteins, including neurotrophins (BDNF, NT4/5), heat shock proteins (Hsp70,

HSP27) and SIRT1 in peripheral blood mononuclear cells in the development of multiple sclerosis.

In my opinion the research results indicate expression and correlations of the factors- what can be connected with MS pathogenesis. This paragraph needs to be specified.

 The text should be carefully revised.

The Discussion is too long.

Author Response

Response to Reviewer 2 Comments 

Point 1: The Introduction is too long.

Introdactions has been shortened to 1207 words (102 lines).

Point 2: 

 2.2.2. Statistical Analysis: The information  if the distribution is normal or not  (probably not because these are human samples) should be included.

In the paragraph Statistical analysis in line 201 we added information about the type of distribution.

“Distribution of variables was assessed by the Shapiro–Wilk test. Statistical analysis of differences between the groups of data was carried out using the Mann-Whitney U-test (for non-normal distribution).”

Point 3:

 Line 225-227, last sentence- should be included to Figure legend.

The sentence was included to Figure legend

 "Expression levels were determined by real-time PCR (see Methods). Statistical analysis of differences between the groups of data was carried out using the Mann-Whitney U-test."

Point 4:

Line 327-329, authors write: In our study, we assessed the effect of microRNAs on the expression level of neuroprotective proteins, including neurotrophins (BDNF, NT4/5), heat shock proteins (Hsp70, HSP27) and SIRT1 in peripheral blood mononuclear cells in the development of multiple sclerosis.

“In our study, we assessed the role of microRNAs in regulating the expression level of proteins with neuroprotective properties, including neurotrophins, heat shock proteins and sirtuins, in the development of multiple sclerosis.” (line 312)

Point 5:

The Discussion is too long.

The Discussion has been shortened to 1308 words.

Best regards

D.Piotrzkowska

Reviewer 3 Report

In this original article (biology-1230400), the authors investigated the effect of microRNAs on the expression level of neuroprotective proteins, including neurotrophins (BDNF, NT4/5), heat shock proteins (HSP70, HSP27) and sirtuin (SIRT1) in peripheral blood mononuclear cells in the development of multiple sclerosis. Some concerns are listed as below:

  1. The sample size is low (13 RRMS and 15 SPMS) in this study. This is a major concern.
  2. In lines 43 and 44, ‘SM’ should be ‘MS’.
  3. In line 53, ‘multiple sclerosis’ should be ‘MS’. The same mistake was also noted in following text. Please double check throughout the manuscript.
  4. In line 73, ‘microRNA’ should be ‘miRNAs’. Please double check throughout the manuscript.
  5. It is not clear for readers why and how several miRNAs were selected in this study.
  6. In the introduction, the role of neurotrophins (BDNF, NT4), SIRT1, and heat shock proteins (HSP27 and HSP27) in the pathogenesis of MS should be discussed.
  7. The hypothesis of this study is lacking.
  8. In table 1, the EDSS score (6) is high. How these patients with RRMS were treated?
  9. I wonder if some DMT drugs may have effect on the expression of microRNAs and neuroprotective proteins.
  10. In table 1, some important information (smoking and ARR) is lacking.
  11. Regarding the statistical analysis, why Pearson, rather than Spearman, was used?
  12. In Figure 1, the expression of BDNF was compared between MS and control. How about the expression of BDNF between RRMS and SPMS?
  13. The authors said that they selected 12 miRNAs that can regulate the expression of BDNF genes (miR-132, miR-182-5p, miR-134). However, the data regarding miR-182-5p is lacking in current manuscript.
  14. The authors concluded that gene and microRNA expression profiling may be a good diagnostic tool in the future for assessing the severity of the disease or estimating survival time. However, available data (no follow-up data) in this study can not support the final conclusion.

Author Response

Odpowiedź do recenzenta 3 komentarze

Punkt 1:

Wielkość próby jest niska (13 RRMS i 15 SPMS) w tym badaniu. To poważny problem.

Nasze badania przeprowadziliÅ›my na grupie 33 pacjentów ze stwardnieniem rozsianym oraz 28 pacjentów z grupy kontrolnej. ZakÅ‚adamy, że tak duża liczba klinicznie reprezentatywnych pacjentów z SM pozwoliÅ‚a na wyciÄ…gniÄ™cie istotnego wniosku wynikajÄ…cego z molekularnych szlaków biogenezy miRNA. Dlatego naszym celem byÅ‚a ocena profilu ekspresji badanych genów i miRNA, ale nie badania populacyjne. Ponadto uzyskane przez nas wyniki wykazujÄ… statystycznÄ… siÅ‚Ä™ istotnoÅ›ci. 

Punkt 2:

W wierszach 43 i 44 „SM” powinno być „MS”.

PoprawiliÅ›my wszystkie bÅ‚Ä™dy redakcyjne. 

Punkt 3:

In line 53, ‘multiple sclerosis’ should be ‘MS’. The same mistake was also noted in following text. Please double check throughout the manuscript.

We corrected all editorial errors.

Point 4:

In line 73, ‘microRNA’ should be ‘miRNAs’. Please double check throughout the manuscript.

We corrected all editorial errors. (83 line)

Point 5:

 It is not clear for readers why and how several miRNAs were selected in this study.

We added this information to the Introduction. (line 89)

“Using software (miRNApath, miRTarBase) and literature data, we selected 12 miRNAs that can regulate the expression of BDNF genes (miR-132, miR-182-5p, miR-134),  NT4/5 (miR-21-5p), SIRT1 (miR-132-3p, miR-34a, miR-34c), HSP70 (miR-378-3p, miR-181b-5p), and HSP27 (miR-577, miR-17-5p).”

Point 6:

In the introduction, the role of neurotrophins (BDNF, NT4), SIRT1, and heat shock proteins (HSP27 and HSP27) in the pathogenesis of MS should be discussed.

Introdactions has been shortened to 1187 words (101 lines).

All text "Introdactions" has been revised and reduced in accordance with the guidelines Reviewer 2. In the introduction, we discussed the role of neurotrophins, SIRT1 and heat shock proteins in the pathogenesis of MS.

Point 7:

The hypothesis of this study is lacking.

In  our study, we assessed the effect of miRNAs on the expression level of neuroprotective proteins, including neurotrophins (BDNF, NT4/5), heat shock proteins (Hsp70, HSP27) and SIRT1 in the development of multiple sclerosis. We postulate, that dysregulation of miRNA levels and the resulting changes in target mRNA / protein expression levels could contribute to development of multiple sclerosis.

Point 8, 9, 10:

  1. In table 1, the EDSS score (6) is high. How these patients with RRMS were treated?
  2. I wonder if some DMT drugs may have effect on the expression of microRNAs and neuroprotective proteins.
  3. In table 1, some important information (smoking and ARR) is lacking.

RR-MS patients were in the remission phase (stable) over 2 years without any attacks, exacerbation or steroid treatment, therefore we did not include - annualized relapse rates (ARRs) in patients characteristics. Moreover RR-MS patients had long time of disease duration 9,7+\-3,2 in that stage of disease, relapses happens less often.   All patients had not received any immunomodulatory therapy for 3 months prior to blood withdrawal. They also, were non-smoking persons for over 1 year.

Point 11:

Regarding the statistical analysis, why Pearson, rather than Spearman, was used?

There was an error in the text, we used Spearman for statistical analysis. Line 206

Point 12:

In Figure 1, the expression of BDNF was compared between MS and control. How about the expression of BDNF between RRMS and SPMS?

Nie zaobserwowaliÅ›my różnicy miÄ™dzy poziomem ekspresji BDNF u pacjentów z SPMS i RRMS.

Punkt 13:

Autorzy powiedzieli, że wyselekcjonowali 12 miRNA, które mogÄ… regulować ekspresjÄ™ genów BDNF (miR-132, miR-182-5p, miR-134). Brakuje jednak danych dotyczÄ…cych miR-182-5p w aktualnym rÄ™kopisie.

Linia 240

Punkt 14:

Autorzy doszli do wniosku, że profilowanie ekspresji genów i mikroRNA może być w przyszÅ‚oÅ›ci dobrym narzÄ™dziem diagnostycznym do oceny ciężkoÅ›ci choroby lub szacowania czasu przeżycia. Jednak dostÄ™pne dane (brak danych z obserwacji) w tym badaniu nie mogÄ… poprzeć ostatecznego wniosku.

Linia 422

Proszę zobaczyć załącznik

Z poważaniem

D.Piotrzkowskiej

Round 2

Reviewer 3 Report

The authors have addressed my concerns.